# Effects of an Alternating Magnetic Field/Ag Multi-Alloying Combined Solidification Process on Cu–14Fe Alloy

**DOI:** 10.3390/ma11122501

**Published:** 2018-12-09

**Authors:** Jin Zou, De-Ping Lu, Ke-Ming Liu, Qing-Feng Fu

**Affiliations:** 1Jiangxi Key Laboratory for Advanced Copper and Tungsten Materials, Jiangxi Academy of Sciences, Nanchang 330029, China; fuqingfeng@jxas.ac.cn; 2Jiangxi Key Laboratory for Precision Drive and Control, Nanchang Institute of Technology, Nanchang 330099, China; jokeliu@sina.com

**Keywords:** Cu–Fe alloy, Ag multi-alloying, combined process, solid solution

## Abstract

An alternating magnetic field (AMF)/Ag multi-alloying combined process was applied to the solidification of Cu–14Fe alloy to study its effects on the microstructure and properties of the resulting samples. The applied AMF and Ag multi-alloying had positive effects on the refinement of the primary Fe phase and precipitation of Fe solute atoms, respectively. These results indicated that the combined AMF/Ag multi-alloying process was effective to improve the distribution of the primary Fe phase and reduce the Fe content of the Cu matrix, which increased the conductivity of the alloy. The application of the combined AMF/Ag multi-alloying process to the solidification of Cu–Fe alloy provided samples with improved comprehensive properties compared with those of samples solidified using a single process (AMF or Ag multi-alloying).

## 1. Introduction

High-strength and high-conductivity deformation-processed Cu-based in situ composites exhibit favorable physical and mechanical properties. Such composites not only have excellent electrical and thermal conductivity but also possess high strength and plasticity, which means they are attractive for applications in the fields of electronics, vehicles, metallurgy, and energy [1,2,3]. Recent studies have shown that Fe is an appropriate strengthening phase for Cu-based composites. Cu–Fe deformation-processed in situ composites are considered to be of great industrial application value because of their excellent mechanical properties and low production cost. However, the insufficient precipitation of Fe solute atoms in the Cu matrix during the solidification process greatly limits the comprehensive properties of Cu–Fe composites [4]. It is important to determine how to inhibit Fe atom dissolution in Cu crystals and promote Fe solute atom precipitation from the Cu matrix to minimize the harmful effects of Fe atoms on the conductivity of the Cu matrix.

Researchers have introduced multi-alloying in Cu–Fe alloys to accelerate the precipitation of Fe during the solidification process by addition of micro-alloy elements, thus inhibiting the dissolution of Fe atoms in the Cu matrix. Ag has been widely studied as a third component to lower the solid solubility of Fe in Cu, and improve the strength of the material without markedly worsening the conductivity [2,5,6,7,8]. Meanwhile, because Fe atoms diffuse relatively slowly from a supersaturated Cu solid solution at room temperature, intermediate heat treatment and final aging treatment are often used to promote precipitation of Fe atoms [9,10,11]. However, these treatments cause grain coarsening and recovery, thus decreasing the strength of the resulting materials. We previously found that the application of an alternating magnetic field (AMF) during the solidification process of Cu–Fe alloys markedly refined Fe grains and effectively reduced the content of Fe solute atoms in the Cu matrix, which was conducive to improve the comprehensive properties of alloys [12,13]. Based on the previous studies, here an AMF/Ag multi-alloy combined process is established. Focusing on Cu–14Fe alloy in this work, the influence of the combined solidification process on microstructure is examined, and the solute distribution and material properties of the obtained samples are evaluated. The aims of this work were (i) to understand the influence of Ag as a multi-alloy addition, (ii) to determine the significance of the AMF/Ag multi-alloy combined process on Cu–14Fe alloy, and (iii) to develop a new method in the research and development of Cu–Fe deformation-processed in situ composites.

## 2. Experimental

The raw materials were electrolytic copper (99.96 wt %), pure iron (99.94 wt %), and pure silver (99.95 wt %). Experimental alloys (Cu–14Fe, Cu–14Fe–0.1Ag, Cu–14Fe–0.5Ag, and Cu–14Fe–1Ag) were melted in a magnesia crucible (Dashiqiao crucible factory, Yingkou, China with a medium-frequency induction furnace (Jinzhou Institute of Metallurgical Technology, Jinzhou, China) possessing a rated power of 35kw, and were then solidified in a graphite crystallizer (diameter *Φ*112 mm) installed in a magnetic field generating system. The temperature of the melt was measured in real time using an infrared thermometer (IRP20, OPTRIS, Berlin, Germany). The solidification experiment was carried out in an AMF generator (IDEA Electric CO.LTD, Shijiazhuang, China) possessing a magnetic flux density of 30mT and field frequency of 26 Hz.

The ingots were initially cut longitudinally along the diameter and the smaller-sized samples were prepared by a wire-electrode cutting to 10 × 10 mm^2^. The samples for investigating the microstructure were polished and etched using a solution of FeCl_3_:HCl:H_2_O = 1:2:20 and were subsequently observed with an optical microscope (OM; DMI3000M, Leica Microsystems, Wetzlar, Germany). The Fe grain size distributions were measured from the OM images using Image-Pro Plus software (6.0, Media Cybernetics, Rockville, MD, USA). Element distribution was analyzed using a scanning electron microscope (SEM; Quanta 200, FEI, Hillsboro, OR, USA) and energy dispersion spectroscopy (EDS; Oxford, UK). The phase structure of the samples was determined via X-ray diffraction (XRD; D8-Advance, Bruker, Billerica, MA, USA) using a tube voltage of 40 kV and a CuKα wavelength of 1.5418 Å. The microhardness was measured using a Vickers hardness tester (Model: HXS–1000, Shanghai Taiming Optical Instrument, Shanghai, China) at a load of 50 g and duration time of 15 s. The conductivity was measured using an eddy conductivity meter (Model: FD-102, Xiamen First Electronic Technology, Xiamen, China).

## 3. Results and Discussion

### 3.1. Microstructure

Figure 1 presents the microstructures of Cu–14Fe and Cu–14Fe–0.1Ag alloys produced by normal solidification. Copper and iron do not form alloy phase during solidification, the typical microstructure of the Cu–14Fe alloys consisted of primary Fe dendrites distributed in the Cu matrix with random orientations. The addition of 0.1 wt% Ag affected the growth of Fe dendrites, decreasing the average sizes of the irregularly shaped grains and secondary dendritic arms. Figure 2 depicts the microstructures of Cu–14Fe alloys with various Ag contents produced under an AMF. The application of an AMF helped to refine and disperse Fe dendrites in the Cu matrix. With increasing Ag content, the size of the Fe grains in the samples was further refined.

The average size and size distribution of the Fe grains in samples produced under various solidification conditions are listed in Table 1 and shown in Figure 3, respectively. As shown in Figure 3, the size distribution of the Fe grains changed markedly with the combined solidification process. Compared with that of the sample obtained by normal solidification, the proportion of grain sizes smaller than 10 µm increased and the proportion of grains larger than 30 µm decreased gradually. In contrast to the combined solidification process, the variation of Ag content in the Cu–14Fe alloy had only a small effect on the average size of the Fe grains.

Addition of a trace amount of Ag to the Cu–Fe alloy had a certain inhibition effect on the size of the Fe grains formed. Similar results have been found in the study of Cu–Nb alloy [14] and Cu–Cr alloy [15]. The size of the primary-phase dendrites in the as-cast Cu base alloy is obviously refined through the action of Ag. Gao et al. [16] reported that Ag can behave as a modifier that can promote nucleation, refining and spheroidizing of the Fe dendrites during the solidification process. In this study, we found that a trace amount of Ag refined the size of the Fe grains but did not obviously make the Fe grains become spherical.

The application of an AMF in the solidification process helped to promote the Fe nucleation and decrease the size of the Fe grains. However, increasing the Ag content had no obvious effect on grain size refinement in the combined process. This may be because the application of an AMF already strongly promotes the refinement of the Fe grains, and the refinement of Fe grains is not a superposition of the effect of the AMF and that of Ag; therefore, the effect of Ag multi-alloying on grain refinement has been covered in the combined process.

### 3.2. Solution Distribution

Figure 4 presents the Fe content of the Cu matrix for samples produced with the addition of Ag in the combined process. The Fe content of the Cu matrix includes the Fe atoms dissolved in the Cu grains and the secondary Fe particles precipitated from the Cu grains during the solidification process, which can reflect the solid solubility of Fe in Cu at the solidus. As shown in Figure 4, the addition of Ag helps to reduce the Fe content of the Cu matrix. With increasing Ag content, the Fe content of the Cu matrix decreases, and the change gradually lessens with increasing Ag content. Wang et al. [17] presented a first-principles calculation that indicated that Ag atoms are more soluble than Fe atoms in Cu. Thus, Ag atoms can occupy Cu lattice positions before Fe atoms, which inhibit Fe atoms from dissolving in Cu, lowering the solid solubility of Fe in Cu.

Fe and Ag have different crystal structures at room temperature (body-centered cubic and face-centered cubic, respectively), and the difference between their atomic radii is 16.9%, which does not meet the requirement for anatomic radius difference of less than 15% in substitutional solid solutions. Therefore, Ag and Fe are almost insoluble in each other, and Ag is completely in solid solution in a Cu matrix.

Figure 5 presents the XRD patterns of the Cu–14Fe alloys obtained by the combined process. Because the Ag atoms are almost in a solid solution in the Cu crystal, no characteristic Ag peaks were observed. However, the characteristic Cu peaks were broadened and shifted as a result of the formation of a solid solution with Ag. Figure 5b shows the position of the Cu(111) diffraction peak for the samples. Compared with that for Cu–14Fe alloy, the diffraction peak shifted to a lower angle upon the addition of Ag. The peak shift increased with the content of Ag, which was caused by the change in the lattice constant of the sample. The XRD behavior of a crystal follows Bragg’s law: (1)2d sinθ=nλ
where *d* is the lattice constant, *θ* is the incident angle, *λ* is the wavelength of the X-ray beam, and *n* is the diffraction series. Thus, the decrease of *θ* indicates the increase of *d*. The atomic radius of Ag is larger than that of Cu, so the Ag solid solution in the Cu crystal will increase both the lattice constant of the Cu crystal and the lattice distortion. The higher the content of Ag atoms, the larger the lattice constant of the Cu crystals, and the smaller the *θ*. Therefore, the shift of the Cu diffraction peak to a lower angle indicates the increasing Ag content of the Cu matrix.

### 3.3. Microhardness

Figure 6 presents the hardness of the Cu–14Fe alloys produced by the combined process. With increasing Ag content, the average hardness of the samples rose slightly. However, considering the measured standard deviation, the effect of Ag content on the hardness of the samples was not obvious.

According to the strengthening mechanism of the Cu matrix in Cu–Fe alloy, the hardness of the sample depends on both solution strengthening and precipitation strengthening [18]. The solution of Ag atoms in the Cu matrix is beneficial to improve the solution strengthening effect of the Cu matrix, and Ag is completely soluble in the Cu matrix. Therefore, the strengthening model of the Cu matrix in the Cu–Fe–Ag alloy samples can be expressed as: (2)$${\mathrm{HV}}_{\mathrm{matrix}}{=(f}_{\mathrm{Fe}-\varepsilon }\cdot {\mathrm{HV}}_{\mathrm{Fe}-\varepsilon }{+f}_{\mathrm{Ag}-\varepsilon }\cdot {\mathrm{HV}}_{\mathrm{Ag}-\varepsilon }{+f}_{\mathrm{Fe}-\mathrm{pre}}\cdot {\mathrm{HV}}_{\mathrm{Fe}}{)+\mathrm{kf}}_{\mathrm{Fe}-\mathrm{pre}}^{1/2}$$
where $f_{\mathrm{Fe}-\varepsilon }$, $f_{\mathrm{Ag}-\varepsilon }$, and $f_{\mathrm{Fe}-\mathrm{pre}}$ are the volume fractions of the Cu(Fe) solid solution, Cu(Ag) solid solution, and Fe precipitation particles, respectively; and, ${\mathrm{HV}}_{\mathrm{Fe}-\varepsilon }$, ${\mathrm{HV}}_{\mathrm{Ag}-\varepsilon }$, and ${\mathrm{HV}}_{\mathrm{Fe}}$ are the hardness of the Cu(Fe) solid solution, Cu(Ag) solid solution, and Fe precipitation particles, respectively.

Because the Ag atoms can dissolve in the Cu crystal prior to Fe and thus inhibit the dissolution of Fe atoms in Cu, the addition of Ag to the Cu–Fe alloy increases $f_{\mathrm{Ag}-\varepsilon }$ and decreases $f_{\mathrm{Fe}-\varepsilon }$. The atomic radii of Cu, Fe, and Ag atoms are 128, 124, and 144 pm, respectively, so both Cu–Fe and Cu–Ag can form substitutional solid solutions. Because the difference between the radius of Cu and Ag atoms is larger than that between Cu and Fe atoms, the lattice distortion in the Cu crystal induced by solute Ag atoms is larger than that caused by solute Fe atoms. This means that the solution strengthening effect of the Cu(Ag) solid solution is greater than that of the Cu–(Fe) solid solution; i.e., the addition of Ag can increase the hardness of the Cu matrix to a greater extent than the addition of Fe.

Although the dissolution of Ag atoms can improve the solution strengthening of the Cu matrix, it also promotes the precipitation of the Fe atoms, thus lowering the solution strengthening of the Fe atoms. A trace amount of Ag used in our experiments had little effect on the dissolved atoms in the Cu matrix. Therefore, the hardness of the samples did not change obviously under the combined actions of the solid solution and precipitation.

### 3.4. Conductivity

Figure 7 presents the conductivity of the Cu–14Fe alloy samples produced using the Ag multi-alloying and combined process. Ag multi-alloying raised the conductivity of Cu–14Fe alloy which evaluated with %IACS (International Annealed Copper Standard). The conductivity of the alloy was further raised when it was produced using the combined process.

Figure 8 shows the conductivity of Cu–14Fe alloy samples with various Ag contents produced by the combined process. Increasing the Ag content of the alloy markedly improved its conductivity for the samples produced by the combined process; however, the increase of conductivity weakened with increasing Ag content. The conductivity of the alloy solidified by the A-M1 (Code) process was about 33% higher than that of the sample obtained by N-0 (Code). Impurity scattering in the Cu matrix is mainly caused by Fe atoms in the Cu solid solution. Although most Fe atoms precipitated gradually during the cooling process, a trace amount of Fe still remained in the Cu matrix as solid solution atoms. The diffusion of Fe atoms in the Cu matrix is slow and they hardly precipitate at room temperature. The resistivity of the Cu matrix increased to 9.2 μΩ·cm when it contained 1 wt % Fe solid solution atoms, and the precipitated Fe particles have little effect on the conductivity of the Cu matrix compared with Fe atoms in the solid solution [19]. It is assumed that the decrease of Fe content in the Cu matrix is related to the dissolved Fe atoms in the Cu matrix rather than the precipitated Fe particles. Moreover, the effect of Ag atoms, impurities, and the Fe-precipitated particles on the resistivity is ignored. The influence of dissolved Fe atoms Δ*x* on the resistivity of the Cu matrix Δ*ρ* can be expressed as follows:(3)Δρ=Δx·9.2(μΩ·cm)

Therefore, on the basis of the change of Fe content in the Cu matrix (Figure 4), the effect of Fe content in the Cu matrix on conductivity can be calculated as follows: (4)$$\varphi =\frac{1.7241}{\rho -\Delta \rho }\times 100\%\mathrm{IACS}$$

The calculated conductivity values for samples solidified by the combined process are also plotted in Figure 8. The calculated values agree well with the measured values and their variation trends are consistent. The addition of Ag can inhibit dissolution of Fe in Cu and decrease the content of Fe atoms in the Cu matrix, which helps to lower the impurity scattering resistance of the Cu matrix. Because the primary Fe grains and Fe precipitated particles have little influence on the conductivity of the samples, this also verified that the change of Fe content in the Cu matrix is mainly that of the dissolved Fe atoms in the Cu matrix. The combined process can promote the precipitation of Fe atoms in the Cu matrix.

## 4. Conclusions

(1) The addition of a trace amount of Ag to Cu–Fe alloy helped to refine primary Fe grains during the solidification process and promote uniform distribution of the Fe grains. The higher the Ag content, the greater the Fe grains were refined. The effect of refining and dispersing of the Fe grains was larger for samples produced by the AMF/Ag multi-alloying combined process than for those produced with AMF or Ag multi-alloying alone. 

(2) The AMF/Ag multi-alloying combined process markedly decreased the Fe content of the Cu matrix. The Fe content of the Cu matrix decreased gradually with increasing Ag content and this change gradually weakened with rising Ag content.

(3) The decrease of Fe content in the Cu matrix caused by the combined process was mainly related to the decrease of the dissolved Fe atoms in the Cu matrix. The decreased content of dissolved Fe atoms in Cu helped to lower the impurity scattering resistance of the Cu matrix, and thus the conductivity of the alloy was improved.

## Figures and Tables

**Figure 1 materials-11-02501-f001:**
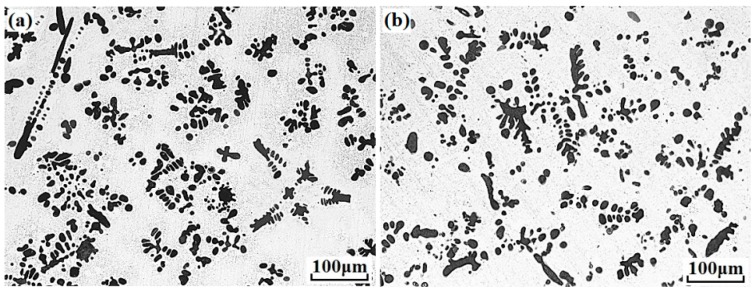
Microstructures of alloys with normal solidification: (**a**) Cu–14Fe; (**b**) Cu–14Fe–0.1Ag.

**Figure 2 materials-11-02501-f002:**
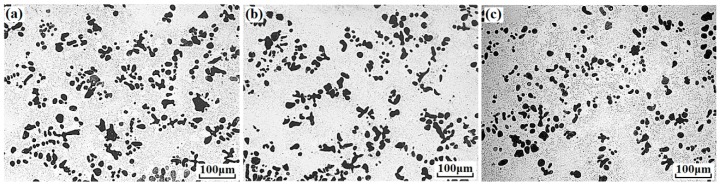
Microstructures of alloys solidified with alternating magnetic field (AMF): (**a**) Cu–14Fe–0.1Ag; (**b**) Cu–14Fe–0.5Ag; (**c**) Cu–14Fe–1Ag.

**Figure 3 materials-11-02501-f003:**
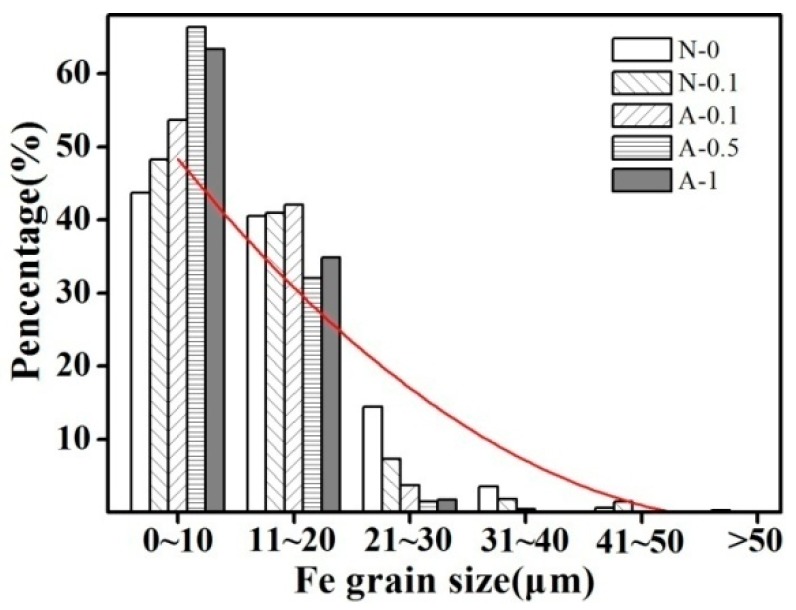
Fe grain size distributions of the various solidification processes.

**Figure 4 materials-11-02501-f004:**
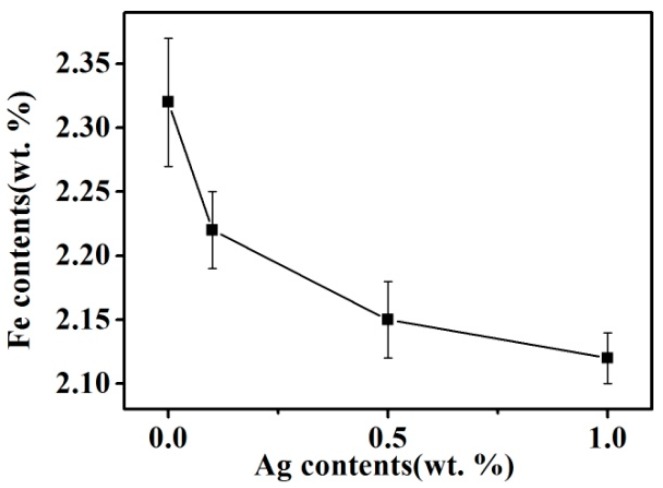
Fe content in Cu matrix of alloys solidified with the combined process.

**Figure 5 materials-11-02501-f005:**
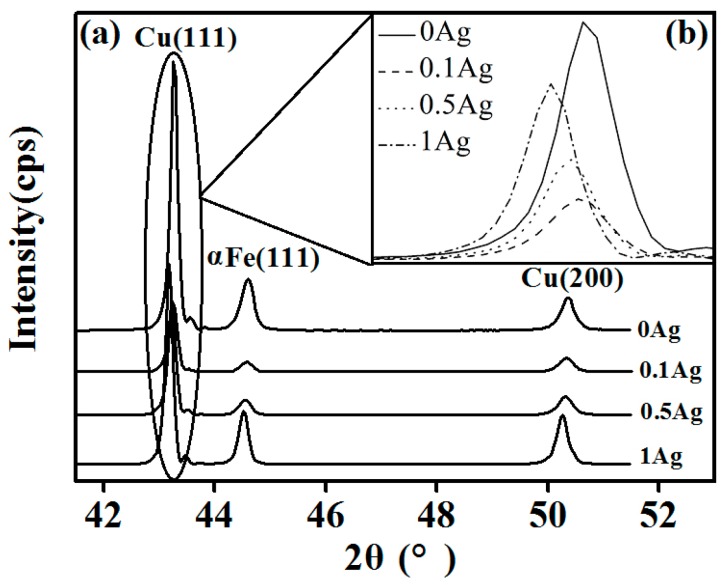
XRD spectra: (**a**) Cu–Fe–xAg alloy (x = 0, 0.1, 0.5, 1); (**b**) Cu(111) diffraction peak.

**Figure 6 materials-11-02501-f006:**
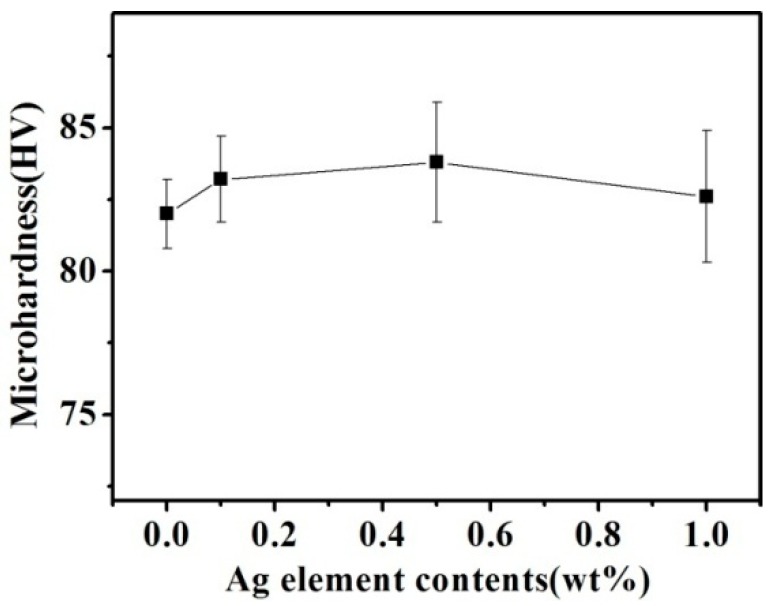
Cu matrix hardness of Cu–14Fe alloys solidified with the combined process.

**Figure 7 materials-11-02501-f007:**
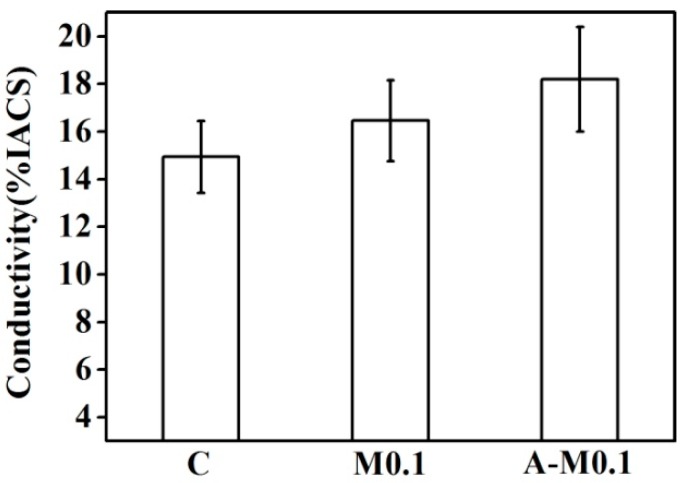
Conductivity of alloys solidified with the Ag multi-alloying and combined process.

**Figure 8 materials-11-02501-f008:**
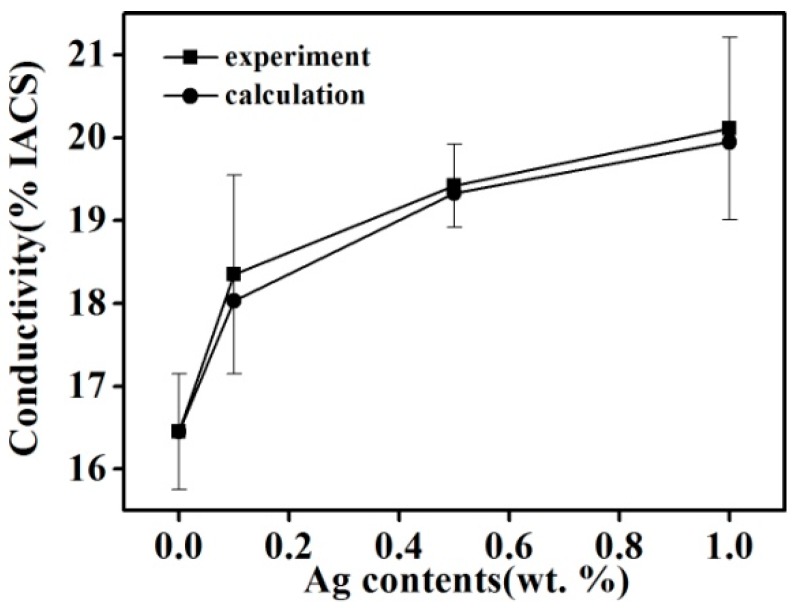
Experimental and calculated values of the conductivity of Cu–14Fe–xAg alloy (x = 0, 0.1, 0.5, 1).

**Table 1 materials-11-02501-t001:** Average Fe grain sizes of the various solidification processes.

Code	Alloy	Solidification Process	Average Fe Grain Size (μm)
N-0	Cu–14Fe	Normal	13.3
N-0.1	Cu–14Fe–0.1Ag		12.3
A-0.1	Cu–14Fe–0.1Ag	AMF	9.99
A-0.5	Cu–14Fe–0.5Ag		9
A-1	Cu–14Fe–1Ag		9.36

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
