# Peer review of "Effects of an Alternating Magnetic Field/Ag Multi-Alloying Combined Solidification Process on Cu–14Fe Alloy"

_materials, 2018, doi:10.3390/ma11122501_

Round 1

Reviewer 1 Report

The manuscript entitled: 'Effects of an alternating magnetic field/Ag multi-alloying combined solidification process on Cu-14Fe alloy' focuses on modifying the microstructure of the Cu-Fe alloy with the minor addition of Ag. I have the following minor comments:

 - Even though the addition of Ag is a kind of motivation, the motivation of the present work has to be emphasized and should be clear

 - All data points should accompany with error bars

 - English language should be polished

 - Typos should be checked carefully

Author Response

Response to Reviewer  Comments

Point 1: Even though the addition of Ag is a kind of motivation, the motivation of the present work has to be emphasized and should be clear.

Response 1: We have listed the aims of this work in manuscript.

Point 2: All data points should accompany with error bars

Response 2: We have revised the Figures.

Point 3: English language should be polished

Response 3: We have requested a native English expert to revise the language

Point 4: Typos should be checked carefully

Response 4: We have revised the manuscript again and again.

Reviewer 2 Report

Paper can be published after some revisions, such as:

1) Experimental. Please replace "irons" and "coppers" with "iron" and "copper"

2) Experimenal or Results and discussion. Does samples interact with crucible at melting? Please describe this point. If so, please give the characteristics of impurity.

3) Fig 4, axes legend. Please replece it with "Fe content, wt. %" and "Ag content, wt. %".

4) Figs 6 and 8, X axe legend. Please replece it with "Ag content, wt. %".

Author Response

Response to Reviewer  Comments

Point 1: Experimental. Please replace "irons" and "coppers" with "iron" and "copper"

Response 1: We have revised in the manuscript.

Point 2: Experimenal or Results and discussion. Does samples interact with crucible at melting? Please describe this point. If so, please give the characteristics of impurity.

Response 2:  The alloys and crucible do not reract at melting. And we have explained in  Results and discussion.

Point 3: Fig 4, axes legend. Please replece it with "Fe content, wt. %" and "Ag content, wt. %".

Response 3: We have revised the Figure 4.

Point 4: Figs 6 and 8, X axe legend. Please replece it with "Ag content, wt. %".

Response 4: We have revised the Figure 6 and 8.
